# Estimation of Leaf Area Index with a Multi-Channel Spectral Micro-Sensor for Wireless Sensing Networks

**DOI:** 10.3390/s22135048

**Published:** 2022-07-05

**Authors:** Laura Maria Comella, Florian Bregler, Eiko Hager, Markus Anys, Johannes Klueppel, Stefan J. Rupitsch, Christiane Werner, Peter Woias

**Affiliations:** 1Department of Microsystems Engineering—IMTEK, University of Freiburg, 79110 Freiburg, Germany; florianbregler@web.de (F.B.); eiko.baeumker@imtek.uni-freiburg.de (E.H.); jo.klueppel@gmail.com (J.K.); stefan.rupitsch@imtek.uni-freiburg.de (S.J.R.); woias@imtek.uni-freiburg.de (P.W.); 2Hydrology, Faculty of Environment and Natural Resources, University of Freiburg, Friedrichstraße 39, 79085 Freiburg, Germany; markus.anys@hydrology.uni-freiburg.de; 3Ecosystem Physiology, Faculty of Environment and Natural Resources, University of Freiburg, Georges-Koehler-Allee 53/54, 79110 Freiburg, Germany; c.werner@cep.uni-freiburg.de

**Keywords:** wireless sensor network, leaf area index, long term deployment

## Abstract

The leaf area index (LAI) is a key parameter in the context of monitoring the development of tree crowns and plants in general. As parameters such as carbon assimilation, environmental stress on carbon, and the water fluxes within tree canopies are correlated to the leaves surface, this parameter is essential for understanding and modeling ecological processes. However, its continuous monitoring using manual state-of-the-art measurement instruments is still challenging. To address this challenge, we present an innovative sensor concept to obtain the LAI based on the cheap and easy to integrate multi-channel spectral sensor AS7341. Additionally, we present a method for processing and filtering the gathered data, which enables very high accuracy measurements with an nRMSE of only 0.098, compared to the manually-operated state-of-the-art instrument LAI-2200C (LiCor). The sensor that is embedded on a sensor node has been tested in long-term experiments, proving its suitability for continuous deployment over an entire season. It permits the estimation of both the plant area index (PAI) and leaf area index (LAI) and provides the first wireless system that obtains the LAI solely powered by solar cells. Its energy autonomy and wireless connectivity make it suitable for a massive deployment over large areas and at different levels of the tree crown. It may be upgraded to allow the parallel measurement of photosynthetic active radiation (PAR) and light quality, relevant parameters for monitoring processes within tree canopies.

## 1. Introduction

Photosynthetic processes in the leaves of a plant play a significant role for CO_2_ sequestration in the Earth’s ecosystem. The efficiency of photosynthetic light energy conversion is essential to understand and model processes of carbon assimilation, the effects of environmental stress on carbon, and water fluxes within tree canopies [1]. The gathering of accurate and temporally dense data about photosynthesis is still today a challenge for the biologist and ecophysiologist, for scaling up gas exchange from leaf to canopy and ecosystem, and for characterizing the canopy-atmosphere interface [2]. The scientific community has undertaken growing efforts in the last years to tackle these measurement challenges, which has eventually resulted in cheap and compact measuring systems [3].

A key vegetation parameter for carbon and water exchange between the biosphere and the atmosphere (e.g., photosynthesis, respiration, transpiration, and interception of precipitation [4,5,6]) is the leaf area index (LAI). This parameter quantifies the areal density of foliage in vegetation as the ratio between the amount of one-sided leaf area (m^2^) in a canopy and the unit horizontal ground surface area (m^2^). LAI measurements are relevant for the monitoring of vegetation growth and for modeling mass and energy exchange at the canopy-atmosphere interface.

Methods to estimate the LAI can be divided into two categories: destructive and non-destructive [2]. The destructive method consists of harvesting the leaves over a defined area of soil and measuring their area. Alternatively, the LAI can be calculated from the weight of the dried harvested leaves over the respective area. These solutions are time consuming and labor intensive, thus, only practicable for small samples of low vegetation types. Moreover, they do not allow the monitoring of the temporal progression of the parameter as the measurement is only possible at the time of the harvesting of the leaves [7]. A non-destructive way to determine the LAI is possible through the measurement of the transmitted radiation through the canopy. Measurement instruments that are based on this approach are, for example, the LAI-2000/2200(C) (LI-COR Inc., Lincoln, NE, USA), the LP-80 Ceptometer (METER ENVIRONEMENT, Pullman, WA, USA), or the SunScan device (Delta-T Devices Ltd., Burwell, Cambridge, UK) [7,8,9]. However, for the LP-80 and the SunScan, the need for a manual operation drives the cost per area and timespan. Moreover, the measurements are dependent on the skills of the operator. Automatic measurements are possible with the LAI-2000/2200, however the monitoring time is limited by the instrument’s internal memory. In addition, the price of the device limits the number of measuring points. Therefore, the available instruments cannot be used to capture the spatial heterogeneity in large forest areas autonomously over multiple seasons. Alternatively, remote sensing data (e.g., LiDAR) can be utilized [2,10], with the drawbacks that no real-time results are available (minor temporal resolution) and that they are restricted by the spatial resolution of their sensors.

Wireless sensor nodes are a good alternative to overcome the issues of manually operated measurement instruments based on the non-destructive approach. Such wireless nodes consist generally of low power computing devices and sensors, that are capable of collecting, processing, and transmitting data to a cloud. Usually, they are designed for long-time deployment. The advances in technology with an ongoing decrease of energy consumption, size, and costs for sensor node systems will in the near future allow to massively deploy them over larger areas and at different levels of the tree crown, offering a measurement of heterogeneity of the parameter of interest in 3D. Due to their novelty and a lack of readiness level, their potential and impact on ecophysiology have not been yet fully investigated.

A Sensor network for LAI measurement has been developed by Bauer, who, in three publications between 2014 and 2019 [11,12,13], describes an automated measurement system for the application in precision agriculture. The LAI serves here as an indicator for the detection of growth anomalies under different environmental conditions. In [14], Shimojo describes a wireless sensor network for precision agriculture and tests his system in a tomato greenhouse. Both Bauer and Shimojo used the off-the-shelf sensor platform TesloB. Qu also developed a wireless sensor network for measuring LAI. The system was initially validated on a corn field [15] and later in a coniferous forest [16].

The common denominator of the described systems is the use of photodiodes and optical filters to measure the light in a specific frequency range. The goal of this work is to go beyond the state-of-the-art and build up a system that is capable of measuring the LAI by developing a wireless sensor node employing the multi-channel spectral microsensor AS7341 from ams-OSRAM AG. The application of this sensor was previously validated in [17] to measure the photosynthetic active radiation (PAR). In this contribution, we will address challenges for using the AS7341 to measure LAI. The result is an outbreaking sensor concept that allows the simultaneous measurement of both PAR and LAI, which in the best of our knowledge, has not been achieved so far. In particular, (1) we introduce the mathematical concept on which our sensing method in based. (2) We will prove the suitability of the sensing approach through a validation in both laboratory and with a long-term field test. (3) We present a method for processing and filtering the sensor data to derive the dynamic course of the LAI cleaned up from noise due to unstable weather or environmental conditions.

Additional features, which increase the competitiveness of our solution in comparison to the existing ones is the fact that it is powered by solar energy to significantly reduce the maintenance that a battery powered system would imply. Moreover, the system is compact and light-weight enough to also be installed on thin branches. In the following, the developed measurement system will be abbreviated as ILAISS (IMTEK Leaf Area Sensor System).

## 2. Background Theory for LAI Estimation

As in [13,14], we use the Monsi-Saeki model [18] for the calculation of LAI. This model, derived from the Beer-Lambert law, gives the relationship between the LAI and the light intensity above ($I_{A}$) and below ($I_{B}$) the tree crown according to Equation (1).
(1)$$I_{B}=I_{A}\cdot e^{-K\cdot LAI}$$

The variable K is the extinction factor, a parameter that according to Monsi and Saeki changes with the type of vegetation and depends on the inclination angle and transmittance of the leaves. K takes values between 0.4 and 1. In this parameter range K=1 holds in case of a horizontal orientation of the leaves, i.e., their angle against the ground is zero, whereas K=0.4 applies for a completely vertical leaf orientation with an angle of 90° against the ground.

It should be noted, however, that the light intensity above ($I_{A}$) and below ($I_{B}$) the canopy is not only affected by the foliage, but also by non-green elements (such as branches or fruits). Therefore, in the case of trees, the ratio between the light intensity above ($I_{A}$) and below ($I_{B}$) the canopy is proportional to the *PAI* (plant area index), i.e., the ratio of plant-covered area to ground area [10]. Thus, all the instruments that are based on optical sensors such as the LAI-2200C can only measure the *PAI*. Thus, it is more precise to write Equation (1) in the following way: (2)$$I_{B}=I_{A}\cdot e^{-K\cdot PAI}$$

A rearrangement of Equation (2) shows that the *PAI* can be calculated by:(3)$$PAI=-\ln\left(\tau \right)\cdot \frac{1}{K}$$
where τ is the relative intensity that is transmitted through the canopy and is given by:(4)$$\tau =\frac{I_{B}}{I_{A}}$$

*PAI* is derived from the sum of the leaf density (*LAI*) and the stem area index (*SAI*). The *SAI* is the one-sided stem area with respect to ground area, with the so-called stem area including branches and stems. Therefore,
(5)PAI=SAI+LAI

To compensate the influence of *SAI* and to determine *LAI*, given *PAI*, the following formula developed by Zou [19] can be employed:(6)LAI=(1−α)·PAI·Ω
where α is given as and Ω stands for the clumping index.
(7)$$\alpha =\frac{SAI}{PAI}$$

## 3. Materials and Methods

### 3.1. LAI Estimation Method

In this work, to obtain the LAI, PAI is measured initially. To do this, we installed two sensor nodes, one above and one below the tree crown, to measure the light intensities $I_{A}$ and $I_{B}$, respectively. PAI can then be determined by Equation (3). However, the value of the extinction factor K must be known. As its assessment is complicated in practical applications, a regression method is used here to determine a function f(·) that includes both the extinction factor K and the correlation between the output signal and ground truth, resulting in Equation (5).

The data for the ground truth are gathered by parallel and synchronous LAI measurements with state-of-the-art instrument (LAI-2200C, LI-COR Inc., Lincoln, NE, USA). Afterwards, the function f(·) is identified through nonlinear regression. We used a power regression model, which minimizes the difference between the LAI that is measured with the newly developed sensor node and the LAI-2200C.
(8)$$PAI=f\left(-\ln\left(\frac{I_{B}}{I_{A}}\right)\right)$$

It must be also pointed out that the measurement of the *LAI* using the Monsi-Saeki model requires the execution of the measurements under diffuse radiation. Direct light causes the scattering of radiation within the canopy, which leads to considerable measurement error [18].

Due to the continuous and over seasonal measurements of the ILAISS, all of the components of Equation (6) can be determined for deciduous trees: The formerly unknown SAI can be estimated at the end of each season after the tree has lost all its foliage by abscission. As the LAI term in Equation (6) becomes zero at this stage, the sensor directly measures the SAI from the relative intensity. This value can then be used for the LAI estimation of the upcoming season.

### 3.2. System Design

To estimate the LAI, an optical sensor has been developed that is based on the AS7341 (ams-OSRAM AG, Premstaetten, Austria). The sensor is embedded in the energy-autonomous sensor node that is shown in Figure 1, which contains the electronics for data acquisition, processing, and transmission. The AS7341 measures light on eleven different channels consisting of pairs of photodiodes with bandpass filters. A total of eight of them are sensitive in the visible wavelength range between 415 nm and 680 nm. One channel measures light in the near-infrared range and one channel is a silicon photodiode without a filter. For converting the analog input signals, the sensor is also equipped with analog-to-digital converters, the gains of which can be adjusted to control the range of sensitivity.

The sensor chip AS7341 is mounted on a PCB as shown in Figure 2. A hole in the housing lid permits light to reach the sensor. This hole is covered by a diffuser disk of polymethyl methacrylate (PMMA), which ensures a homogenous illumination on the sensor. The housing of the node acts as a shadowing ring, blocking incoming radiation from an angle bigger than 180 degrees to prevent the sensing of radiation reflected by the ground. The glass dome protects the sensor and the circuitry from dirt or water. Figure 1 [17] shows a schematic of the cross-section of the sensing part of the sensor node.

The described sensing unit has been integrated in a light and compact platform, with a total weight of only 46 g, suitable for the deployment on different levels of the canopy (see Figure 1). A cap allows the selection of the field of view of the sensor in such a way that the zenith angle is 73° and the azimuth angle is 90°. This restriction is set in the LAI-2200C for single tree measurements, which is the case for our field test (see Section 3.4).

The sensor node is modularly structured (see Figure 1). One printed circuit board (PCB) at the bottom of the sensor node housing hosts a microcontroller for data acquisition and processing as well as a dedicated hardware module for data transfer using the LoRaWAN protocol. This communication protocol offers a km range data transmission at low power consumption. Both are relevant characteristics to achieve large field deployment at a minimum allowable maintenance. The sensor node is powered through harvested solar energy. The solar cells are all integrated on the top PCB and are directly connected to power management and energy storage in the middle PCB which also carries the light sensor. The antenna protrudes from the bottom, not to hinder the sensor field of view.

### 3.3. Sensor Validation

To validate the sensor concept and characterize its properties, the sensor system was characterized first in the laboratory and afterwards it was tested in a measurement campaign to validate its performance for long-term deployment. For all the experiments, the state-of-the-art instrument LAI-2200C (LI-COR Inc., Lincoln, NE, USA) was used to validate and calibrate the newly-developed sensor node. 

For the characterization, we conducted two experiments: (1) the intensity test and (2) the transmission test. In the intensity test, the radiation intensity was measured with the sensor node and the LAI-2200C when exposed to the same constant radiation. In the transmission test, the transmission of both the sensor node and the LAI-2200C was measured when the sensors were completely covered with a leaf. The aim of these tests is to achieve a better understanding of the functionalities of the optical sensors that are mounted on the LAI-2200C and on the sensor node in order to compare them. Both instruments use optical sensors for the measurement. However, no information is available on the chip as well as the filters that are utilized in the LAI-2200C. Moreover, our newly-developed sensor node exploits the Monsi-Saeki model to calculate the LAI from the transmission measurements, whereas the LAI-2200C is based on the Miller model [20]. Hence, a better understanding of the sensors functionalities would permit the identification of the origin of differences between the LAI measurements done with the two systems. Details about the execution and the results of these experiments will be discussed in Section 3. 

Afterwards the regression function evaluation test was executed. The new sensor node and LAI-2200C were mounted in parallel on a clamp. An operator took measurements of the transmitted light below the canopy and of the light intensity above the canopy. The gathered data were used to calibrate the newly-developed sensor node, thus identifying the calibration function f(·) discussed in Section 3.1.

Finally, we tested the newly-developed sensor concept in a field experiment with deployment. The purpose of this test was to verify the performance and accuracy of the sensor node’s output under real conditions. Therefore, two sensor nodes were installed below and outside the crown for 75 days, during which the course of the LAI before and after leaf abscission could be measured.

### 3.4. Measurement Field Site

The site for the measurement was Freiburg in Breisgau, Germany, on the campus of the Technical Faculty of the Albert-Ludwig-University. The precise location of the measurements for the regression function evaluation and the type of trees are shown in the Table 1 and Figure 3.

For the field experiment with deployment, the sensor node B was installed on Tree 1 of Table 1, by fixing it to a bar (see Figure 4). The reference node A was placed on a glass roof at the position 48.01353° N, 7.83358° E.

### 3.5. Data Processing of Field Experiment with Deployment

The gathered light intensities of the two sensor nodes have to be first filtered and processed to obtain a valid LAI reading. Data which do not fulfill the requirements of the Monsi-Saeki model must be excluded. Moreover, data preprocessing and filtering is necessary to remove noise that is generated from (1) unstable weather not guaranteeing diffuse radiation conditions, (2) measurement inaccuracies below the canopy due to sudden movement of the leaves that is caused by the wind, and (3) technical issues such as missing data. In this section the filters, the preprocessing, and postprocessing methods, shown in Figure 5, will be presented.

#### 3.5.1. Preprocessing

##### Light Intensity Filter

If the radiation intensity is too low or the samples were recorded in complete darkness, there is no reasonable information available for the LAI calculation. Thus, a filter is applied which sorts out all the measurements m(t) coming from every node that is below a defined threshold $S_{I}$ according to: (9)$$m{\left(t\right)}^{\prime }=\{\begin{array}{cc} m\left(t\right) & \mathrm{if}m\left(t\right)\geq S_{I}\\ Ø & \mathrm{if}m\left(t\right)<S_{I} \end{array}$$

The threshold value was defined for the respective sensor nodes in such a way that its value is 10% smaller than the lowest daily mean value of the measurement campaign.

##### Data Aggregation

The two sensor nodes are, by design, not synchronized and transmit data packages independently that include a timestamp. On evaluation, the measurements with the smallest time difference get paired. A minimum threshold $S_{t}$ was defined for the acceptable time interval. If the temporal difference between the measurement’s execution is below the threshold, the measurements will be paired, otherwise they will be discarded according to:(10)$$p\left(t\right)=\{\begin{array}{cc} m_{A}\left(t\right)\cup m_{B}\left(t^{\prime }\right) & if\exists t,t^{\prime }:\left|t-t^{\prime }\right|<S_{t}\\ Ø & \mathrm{elsewhise} \end{array}$$

For the field test, a threshold $S_{t}$ of 60 s was defined.

#### 3.5.2. LAI Calculation

The filtered data of the light intensity above and below the crown were finally utilized to calculate the PAI using Equation (3). We identified the SAI using the mean value of the relative intensity when all the leaves were shed and only branches were left. The LAI was calculated using Equation (7). The clamping index (CI) Ω in Equation (7) was set to one. This index describes the phenomenon of the non-random distribution of leaves in the horizontal plane causing an underestimation of the LAI while using optical measurement methods. The determination of CI using our system is not possible. However, its neglection for the measurement campaign is legitimated since the measurements with the LAI-2200C at the same position showed a CI of Ω>0.99.

#### 3.5.3. Post Processing

##### Diffuse Radiation Filter

To exclude datapoints which were not measured under diffuse radiation, required for Equation (3) to be valid, a filter was introduced. This filter proves the suitability of the data on the basis of the ratio R(t) between the global intensity $I_{G}$ and the diffuse intensity $I_{D}$. A high value of R(t) corresponds to conditions with direct sunlight in the absence of clouds, whereas a low value corresponds to cloudy conditions and, therefore, diffuse light. The implementation follows Equation (11), thus each data pair *p*(*t*) with a ratio R(t) below a certain threshold $T_{R}$ is omitted.
(11)$$p{\left(t\right)}^{\prime }=\{\begin{array}{cc} p\left(t\right) & \mathrm{if}R\left(t\right)\geq T_{R}\\ Ø & \mathrm{elsewhise} \end{array}$$

A threshold of $T_{R}=0.95$ was identified as the most suitable to filter out outliers that were caused by direct light, guaranteeing at the same time that most of the data taken under diffuse conditions remains. The effect of the filter can be seen in Figure 6. Its application leads to the removal of low-frequency outliers.

##### Average Calculation

The last step of post-processing is the calculation of the LAI daily average, which compensates the high frequency noise that is caused, for example, by the wind. Additionally, to the daily mean, we applied a simple moving average filter (SMA) such as that suggested in [16] with a sliding window size of eight days.

## 4. Results

### 4.1. Intensity Test

In this experiment, the relative intensity of the radiation was measured with our sensor node and the LAI-2200C when it was exposed to the same constant radiation. The relative intensity τ(S) with the power level S is calculated as the ratio between the measured radiation intensity at a specific power level and the radiation intensity that was measured when the power source is at the highest power level, i.e.,: (12)$$\tau \left(S\right)=\frac{I\left(S\right)}{I\left(S=10\right)}$$

To ensure the identical alignment of the LAI-2200C and of the sensor node, a clamp was designed (see Figure 7). This allows to minimize the distance between the center point of the LAI-2200C lens and the glass dome of the newly-developed sensor node.

To expose the sensors to a constant light intensity over time, to exclude the influence of unknown external radiation, and to guarantee diffuse radiation during the tests, we performed experiments in the photo box HPB-60D from HAVOX Fotostudio. A polyester fabric was used to obtain diffused radiation. The diffusor fabric was placed on a transparent pane of PMMA and between the light source and the sensors as can be seen in Figure 7.

Measurements with the sensor node and the LAI-2200C were triggered within a time difference of maximum 1 s. The measurements were executed at different power levels of the light source: between power level S=10, which corresponds to the maximal power, and power level S=0 when the light is switched off. Figure 8 shows the relative intensity that was measured with the sensor node and the LAI-2200C for the different power levels of the light source. It must be pointed out that for the sensor node, the average value $\bar{\tau \left(S\right)}$ over the number of sensor channels is plotted.

The relative error per power level is calculated as the ratio between the difference of the relative intensity that was measured with our sensor node and that of the LAI-2200C. The calculated relative errors per power intensity are then averaged according to:(13)$$\bar{err_{rel}}=\frac{1}{N}\overset{N}{\underset{i=0}{\sum }}\frac{\left|\tau_{n}-\tau_{\mathrm{LAI}2200}\right|}{\tau_{\mathrm{LAI}2200}}$$

For the conducted experiment, we obtained as result an $\bar{err_{rel}}$ = 0.01, i.e., the dependence between the power level and the relative error is not evident. The test, thus, shows that under laboratory conditions, the deviation between the relative light intensity measured with the sensor node and the LAI-2200C is negligible. In other words, the measurements of the intensity change under the same variable light conditions correspond very well for both instruments.

### 4.2. Transmission Test

The sensitivity range of the LAI-2200C is between 320 nm and 490 nm [21], whereas for the spectral sensor that is used in the presented sensor node, it lies between 415 nm and 680 nm in the visible range [22]. Since the transmittance and the reflectance of the leaves depend on the wavelength, differences between the transmission that is measured with the LAI-2200C and with the different channels of the newly-developed sensor are expected. To estimate the range of the measurement differences, the transmission of the sensor node and of the LAI-2200C was measured when the sensors are completely covered with a leaf of Norway maple tree. For this purpose, both instruments were equipped with a ring that allows their complete coverture as shown in Figure 9.

The light intensity with leaf cover and without was determined with three measurements each. These three measurement results were averaged and the transmission was obtained from the ratio between the measured light intensity with and without cover. We carried out the described test for three different leaves from the same tree since the transmission is dependent on several parameters, such as the chlorophyll concentration, and the leaves belonging to the same tree can show different transmission levels [23]. Moreover, it must be considered that the water content of the leaves can influence the transmission. As after the cutting, the water content decreases with the time, all the measurements were taken with a constant delay time of 5 min between cutting and measuring.

In Figure 10, the relative transmittance that was measured by different channels of the spectral sensor and the LAI-2200C for the three leaves are displayed. It can be observed that the LAI-2200C measures a transmittance of $\bar{\tau_{\mathrm{LAI}2200}}=$ 0.021, which is within the range of the transmittance that is documented in the literature for this type of a tree [23]. The transmittance that was measured with our sensor node is in general higher for all integrated channels. The sensor node’s channel 3, with a peak sensitivity at 480 nm, shows the smallest transmittance of $\bar{\tau_{node}}=$0.105. The unexpected high transmission level of the sensor node can be explained with the fact that the filters of the spectral sensor are not filtering the NIR (near infrared) light completely. To minimize the difference between the LAI that is measured with the newly-developed sensor node and the LAI 2200C, we have chosen channel 3 for the subsequent experiments.

The transmittance factor of the leaves influences mostly the measurements in the high LAI range. For a lower LAI, the radiation through the gap between the leaves dominates compared to the radiation that is transmitted through the leaves. Therefore, the biggest difference between the LAI measurements of the LAI-2200C and sensor node will be expected in the higher LAI range. Multiple leaves in the path of a radiation ray further reduce the transmittance factor as compared to a single leaf [24] and, thus, have an impact on the LAI measurements. We compensated the deviation of the transmittance factor between the measurement instruments by applying an appropriate regression function.

### 4.3. Regression Function Evaluation

A field experiment was conducted in which light-intensity above and below the canopy was measured with the newly-developed sensor node and the LAI-2200C as a reference. The gathered data were recorded with the aim to correlate the output of the sensor nodes and the LAI-2200C, and thus to determine the function f(·) that is discussed in Section 3.1 with the regression method.

Measurements were taken on three different trees at the following days of the year (doy): 238, 243, 252, 253, 257, 270, and 279 in 2021. These days were chosen because the weather conditions fulfilled the requirement of diffused radiation that was necessary for applying the Monsi-Saeki model. In total, 69 LAI measurements were collected, 19 for Tree 1, 33 for Tree 2, and 17 for Tree 3. The position of the instruments and the number of measurements per single LAI estimation was chosen according to the measurement method for isolated trees given in the instruction manual of the LAI-2200C [21]. The measurement instruments were pointed in the same compass direction for both measurements to ensure consistent conditions. Both measurement instruments were restricted in the azimuth angle to 90 degrees by attaching a view-limiting cap.

Figure 11 depicts the 69 LAI measurements that were collected with both the newly-developed sensor node and the LAI-2200C. The degree of correlation between the measurements that were executed with both instruments was quantitatively specified by a correlation coefficient of $r_{LAInode,LAI\mathrm{LAI}2200}=0.97$. This value proves a strong linear relationship between both instruments. The average relative error that is related to the measured values of the LAI-2200C is 0.216 and the absolute error nRMSE = 0.97.

To determine the optimal regression function based on the measured values, the mean square error (MSE):(14)$$\mathrm{MSE}=\frac{1}{N}\overset{N}{\underset{i=0}{\sum }}{\left(LAI_{\mathrm{LAI}2200}-f\left(LAI_{node}\right)\right)}^{2}$$
between the individual measurements of the two measuring instruments was considered after applying the selected regression function, here a linear and a power function. Figure 11 shows how both the linear and the power curve fit the 69 LAI measurements. First, we investigated the use of the linear function f(x)=mx+b plotted in orange in Figure 11. For m=1.409 and b=−0.299, the linear function is such that the MSE is minimized. With this polynomial function, a fitting error $\bar{err_{rel}}=0.094$ and nRMSE = 0.113 is achieved. However, this function delivers negative output values for input values < 0.3 as shown in Figure 11 and, thus, is not well defined over the whole range.

An alternative function type that fulfills this requirement is:(15)$$f\left(LAI_{node}\right)=a{\left(LAI_{node}\right)}^{r}$$
with a,r∈R, plotted in green in Figure 11. For a=1.023 and r=1.223 the smallest MSE is achieved. In comparison to the linear regression, a smaller relative error is obtained $\bar{err_{rel}}=0.088$, nRMSE = 0.117 is comparable. It follows that by applying this function for calibration, the deviation between the two instruments is reduced approximately to one third. Moreover, the function is well defined over the whole LAI range. Figure 11 shows additionally in black dashes and as reference the curve which would provide the ideal correspondence between the measurements that were executed with the newly-developed sensor and with the LAI-2200C.

The error remaining after the use of the regression function can be caused from a slightly different field of view of the two instruments. This may happen firstly because of the offset position of the sensors, which cannot be avoided if simultaneous measurements are required. Secondly, a minimal difference in the rotation of the sensor cover can change the sensor field of view (FOV). Another reason is that the FOV of the LAI-2200C is divided into five rings to measure at different zenith angles and a weighting according to the Miller Modell is applied [21]. Such weighting is not possible for our sensor node, as the view area is not divided. This has a strong effect especially with an inhomogeneous canopy. These effects cannot be compensated by applying the regression function since the resulting different FOV does not correlate with the LAI value.

Moreover, the bigger absolute difference between the measurements of higher LAI values can be explained with the fact that the relationship between transmittance and LAI is logarithmic. Therefore, a change of the relative intensity for a small value results in a bigger change of the LAI than in the case of higher relative intensities.

### 4.4. Field Experiment with Deployment

To verify the accuracy of the sensor node’s measurements under real conditions, we conducted a measurement campaign. The field experiment lasted 75 days between the 3 September 2021 (doy 246) and the 17 November 2021 (doy 321), in which the system was able to measure the course of the LAI before and after leaf abscission. This period was chosen because in this time of the year, a fast rate of change in the course of LAI is expected at this geographical location. The sensor nodes were configured to acquire data every day between 5 a.m. and 7 p.m. (time in UTC time zone) in a one-minute interval. Such a high measurement frequency, which is not required to follow the course of LAI, was set here to validate the system and the data processing.

For the LAI assessment, two specific positions of the sensor node are required: a sensor node B below the crown and a corresponding reference sensor A above. The sensor node B was mounted on a holder, which permits fine adjustments of the sensor angle. The node was installed at a distance d = 0.4 m from the trunk, a height h = 2.4 m from the ground and it was aligned with the compass direction 185 degrees. The reference node A was placed on a glass roof at 2.35 m from the ground and oriented along the same compass direction of node B. The measured LAI values that were obtained during the field test and filtered with the process that was previously discussed are displayed in Figure 12.

The plot additionally contains the measurements of the LAI that were executed with the state-of-the-art instrument LAI-2200C, to validate the results of the sensor node. Additionally, to illustrate the correlation between the measured LAI and the visual change in the tree crown, a photo series of the considered tree is shown in Figure 13.

The photos were taken on different days of the measurement campaign from the same position in a southerly direction. The photo series starts after the first leaf discoloration and ends at the time of complete leaf loss. In the time period between the beginning of the campaign on doy 246 and doy 280, an average constant LAI of 3.3 can be observed. From doy 280 on, a small decrease of the LAI was observed. This could be attributed to the drop of the seeds of the maple tree.

A strong reduction of the LAI is evident between doy 293 and doy 304, confirmed from the pictures, in which it is observable that on doy 306 most of the leaves were shed. No big changes in the LAI value were observed from this point in time on. The stationarity of the LAI value is confirmed from the pictures, in which no visible difference can be seen after doy 306; individual withered leaves had not fallen off even by the end of the measurement campaign. Based on the measurement values, one can conclude that the measurement system is able to record the dynamic course of the LAI. It allows, for example, to identify the beginning and the end of the leaf abscission.

To validate the results of the field experiment, seven measurements of the LAI were taken in parallel with the LAI-2200C and are shown in the graph with red symbols. The relative errors between the measurements that were taken with our sensor node and the LAI-2200C were calculated and averaged, resulting in RMSE = 0.149, a normalized root means square error nRMSE = 0.098, and a coefficient of determination R^2^ = 0.997. Those factors were used to compare the result of our measurement campaign with other relevant works in the field.

The only measurement campaign in which the same validation method of this work was used is described in [15], in which a RMSE = 0.17 and R^2^ = 0.92 was obtained. The measurement error that was obtained in our campaign is, thus, lower than the one that was obtained in [15] and the coefficient of determination closer to one.

In [12], a sensor network for estimating the LAI was also designed. Even if the validation method is different than the one that is considered in this work, the nRMSE = 0.07 is provided. This value is comparable with our results. Another study is shown in [14], in comparison to which the developed system has better RMSE and coefficient of determination.

It must, however, be pointed out that a good comparison between the studies is complicated as the number of validation measurements differs. A good comparison is possible with [12], as in this work the nRMSE is provided, and with [15], as the used validation method corresponds with the one in this work.

## 5. Discussion

In this contribution, we propose an innovative wireless sensing method, which allows the continuous measurement of LAI in an automatic way. This is a relevant parameter to understand and model ecological processes, which in the proposed form allows the monitoring of heterogeneity in complex ecosystems. However, its continuous observation using traditional assessment methods, when possible, is cost- and time-intensive.

The presented sensing method goes beyond the state-of-the-art as a spectral sensor is used instead of a single photodiode. Thereby, the sensor that was validated here for measuring the LAI is also able to measure PAR and, in general, the radiation distribution under the tree crown.

The newly-developed sensor was initially characterized in the laboratory for understanding its functionalities in comparison with the LAI-2200C and its suitability for measuring the LAI was successfully proven Afterward, we validated the sensor node with measurements in the field that were executed by an operator. Temporally synchronous and spatially parallel measurements were executed using the sensor node and the state-of-the-art instrument LAI-2200C. The obtained measurements were used to calibrate the sensor node. Therefore, a regression function was experimentally defined which minimizes the deviation between the LAI measurements of the LAI-2200C and of the developed sensor node. The measuring system was finally successfully validated in a measurement campaign to evaluate its performance in a real environment.

The second main aspect of the contribution, besides the sensor, was the development of an approach to process and filter the obtained data. Close to standard filtering mechanisms, such as a daily and a multi-daily mean filter, we implemented a diffuse radiation filter to guarantee that the measurements considered fulfill the requirement of the Monsi-Saeki model. The filter reduces the low-frequency outliers and their effect on the signal showing the LAI trajectory.

The campaign revealed the suitability of the measuring system to determine the course of LAI in a high temporal resolution. The obtained LAI trajectory follows the seasonal variation of the tree crown and permits the identification of the beginning and the end of the abscission period. A nRMSE of 0.098 was observed between the measurements that were done with the newly-developed sensor node and the validation measurements that were executed in parallel with the LAI-2200C. The measurement campaign, however, was executed with two sensor nodes only. Hence, the temporal continuity of the measurement system was evaluated but not the three-dimensional heterogeneity of the LAI parameter in the crown. A bigger number of distributed nodes could further improve the sensor accuracy and would help in estimating the clumping. These aspects were not addressed here and will be one of the objectives of following works.

In comparison to other wireless sensor networks that are documented in the literature, the newly-developed sensor performs equally when not better with an nRMSE = 0.098, RMSE = 0.149, and R^2^ = 0.997 (see Tale 2). Its superiority lays additionally in the fact that (1) it is fully powered from a solar cell so no maintenance is required once the node is deployed, and (2) it could allow the parallel measurement of PAR, LAI, and light quality. As shown in Table 2, this is not the case for other sensor nodes being applicable for outdoor monitoring. The use of an optical filter restricts in their case the range of the radiation wavelength allowed as input, making them unsuitable for PAR.

Challenging aspects, which will be addressed in future work with focus on the wireless sensor network, will be prolongation of the sensor node’s operational lifetime through energy optimization and energy awareness. Algorithms will be developed to make the sensor node aware of its energy status and able to forecast its energy income in the near future. Based on its energy awareness, the sensor node will be able to actively tailor its functionality accordingly. To achieve this, the optical sensor will play an important role. It will simultaneously monitor the LAI and the energy income in terms of the input light. The advantage of this solution is that in comparison to algorithms using weather station data, we will have a very specific and local information to run optimized algorithms. Another relevant aspect, which will be further investigated, is the design of a special wireless mesh network for a forest environment.

Moreover, the integration of additional sensors on the node is planned. By embedding additional sensors, it will be possible to gather several types of data continuously and automatically in different levels of the crown, opening new possibilities in terms of modeling of ecophysiological processes and stress tracing. This will additionally improve the measurement performance of the system. The diffuse radiation filter, for example, is based on measurements coming from a weather station that is located approximatively one kilometer away from the location of the measurement campaign. However, such a small distance to the measurement site cannot be guaranteed in any location. A local measurement would improve the filter’s performance. Therefore, a technical solution will be investigated to integrate such measurement on the sensor node, possibly by using the spectral sensor itself.

The developed sensor works ideally in the case of a homogenous tree crown, so a further challenge to be addressed in the sensor’s technical development is its optimization for non-homogenous crowns. This could be achieved by the use of sensors that are able to measure the incoming light at different angles, by using additional optics or by installing multiple regularly spaced sensors to measure the LAI in correspondence with different distances from the trunk. Moreover, algorithms will be developed to enable the sensor to track the change of color of the leaves, for example, to identify a correlation with diseases or pests spread.

## 6. Conclusions

In this contribution, we have presented an innovative sensing concept to measure the LAI by using the spectral sensor AS7341. The sensor is embedded on an energy-autonomous and wireless sensor node, thus generating a novel LAI measurement system called ILAISS. ILAISS can be deployed at remote locations and for a long time, thus being able to acquire data over several seasons in an automatic way. Its energy autonomy, wireless connectivity, and low maintenance effort also make it suitable for a massive deployment over large areas and at different levels of the tree crown, offering a measurement of the heterogeneity of the parameters of interest in 3D. Due to the continuous and over-seasonal measurements of the ILAISS, WAI can be additionally obtained from the measurements in the absence of leaves. The LAI can be obtained by subtracting this value from the PAI, which is measured during the period of foliaged branches. Thus, the newly developed sensor system, unlike many other LAI instruments that are based on optical sensors, enables the determination of both PAI and LAI.

Additional features, which increase the competitiveness of our solution in comparison with the existing ones is the fact that (1) it is powered by solar energy to significantly reduce the maintenance that a battery-powered system would imply, and (2) it is compact and light-weight enough to be installed also on thin branches. The laboratory tests as well as the field experiment have proven its good performance. In comparison to other wireless sensor networks that are documented in literature measuring LAI on trees, the newly-developed sensor with an nRMSE of only 0.098 performs better than the existing ones.

Innovative—and in the knowledge of the authors not documented in literature yet—is the fact that the developed sensor may be upgraded to allow the simultaneous measurement of both the PAR and LAI. In future studies, such a combined sensor node could be used to evaluate the effect of photosynthetically active radiation and, in general, the effect of light in specific ranges of wavelength to the plant growth.

## Figures and Tables

**Figure 1 sensors-22-05048-f001:**
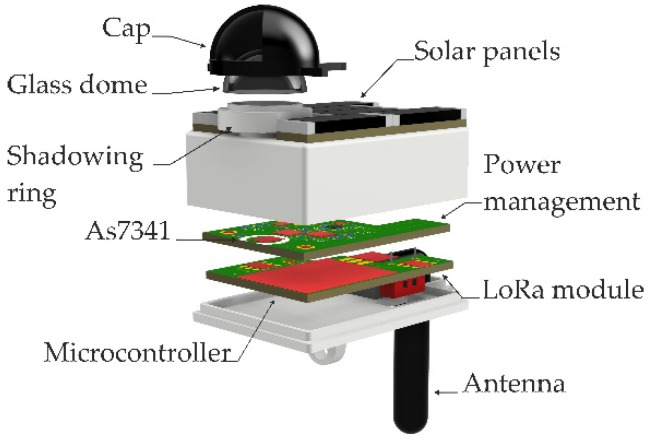
Sensor node of the ILAISS and its components (Rendering).

**Figure 2 sensors-22-05048-f002:**
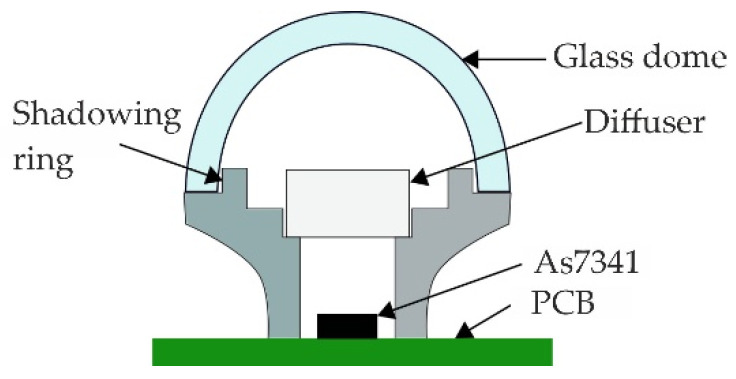
Schematic representation of the cross-section of the sensing part of the sensor node.

**Figure 3 sensors-22-05048-f003:**
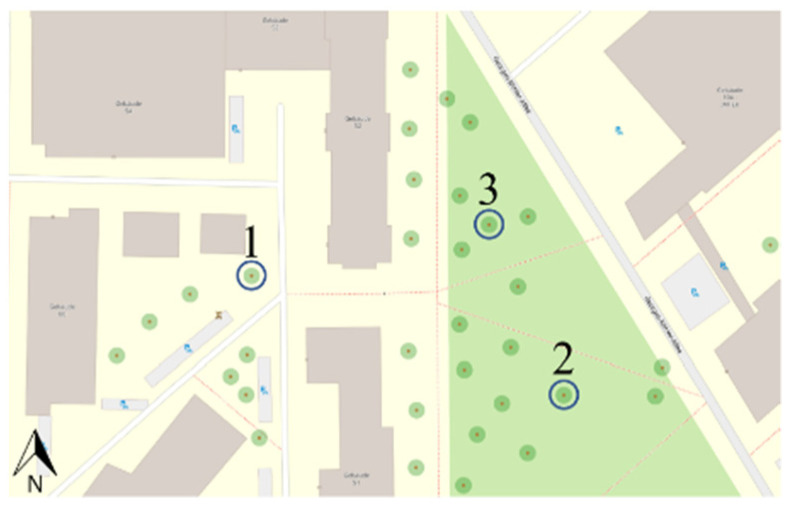
Marked position of the trees that were considered for the experiment, numbered from 1 to 3 (modified map from openstreetmap.org accessed on 10 January 2022).

**Figure 4 sensors-22-05048-f004:**
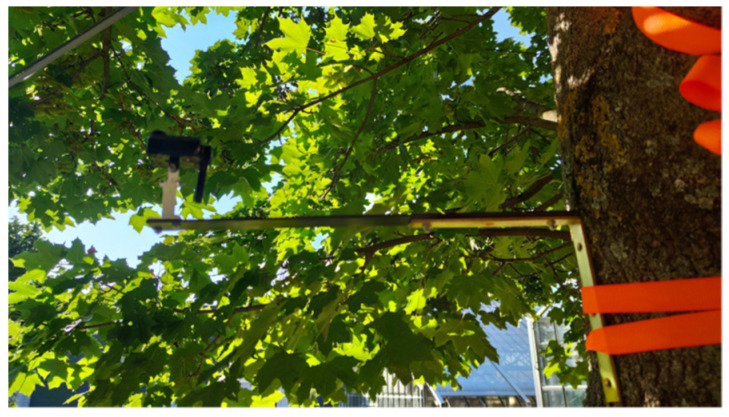
Sensor node B below the canopy of a Norway maple tree for executing continuous and automated LAI measurements.

**Figure 5 sensors-22-05048-f005:**
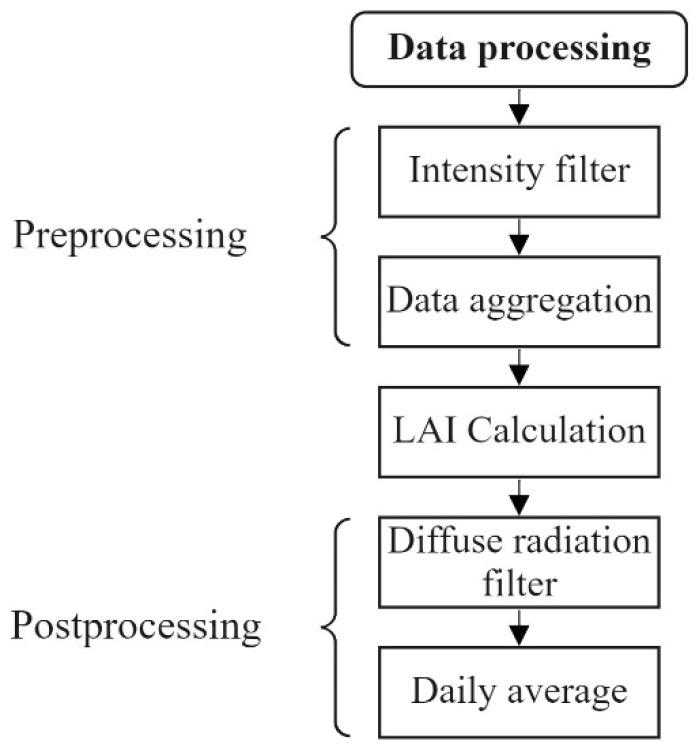
Overview of the developed data processing that was applied on the measurement data of the field experiment.

**Figure 6 sensors-22-05048-f006:**
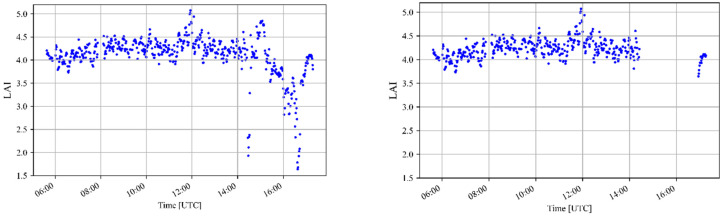
The measured LAI with respect to the daytime during doy 259 before (**left**) and after (**right**) applying the diffuse radiation filter.

**Figure 7 sensors-22-05048-f007:**
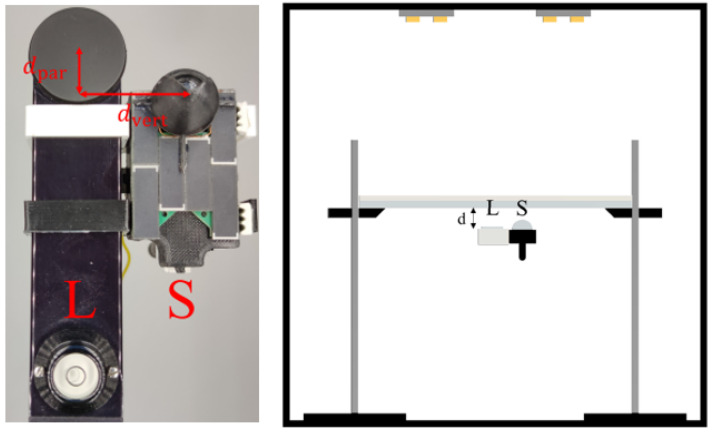
Setup to measure the radiation intensity under laboratory conditions using the LAI-2200C measurement instrument (L) and the sensor node (S).

**Figure 8 sensors-22-05048-f008:**
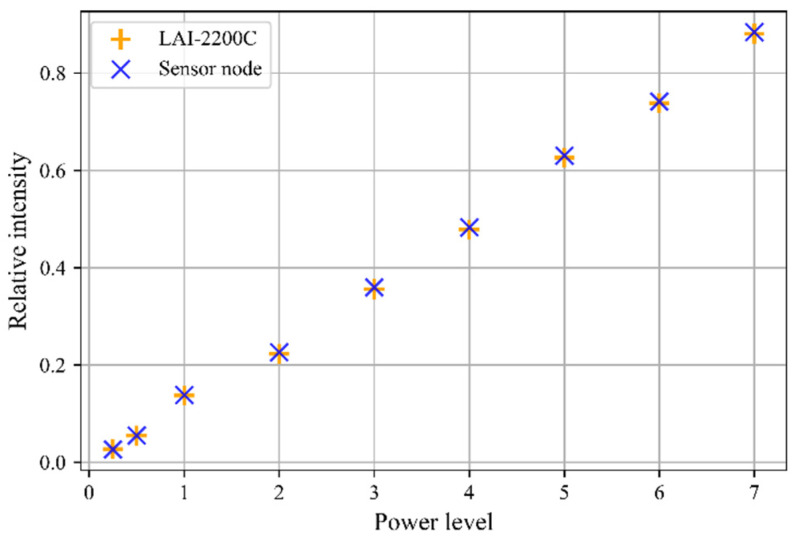
The relative radiation intensity that was measured by the LAI-2200C instrument and the developed sensor node as a function of the power level of the radiation source.

**Figure 9 sensors-22-05048-f009:**
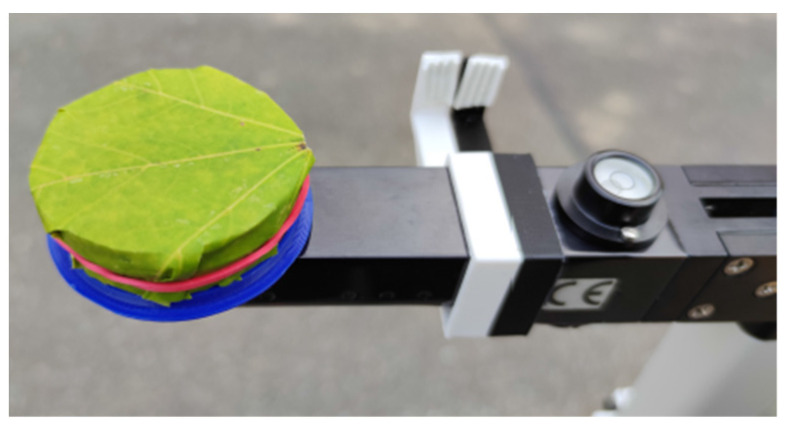
Photo of the setup to measure the transmittance of a single leaf of the Norway maple tree.

**Figure 10 sensors-22-05048-f010:**
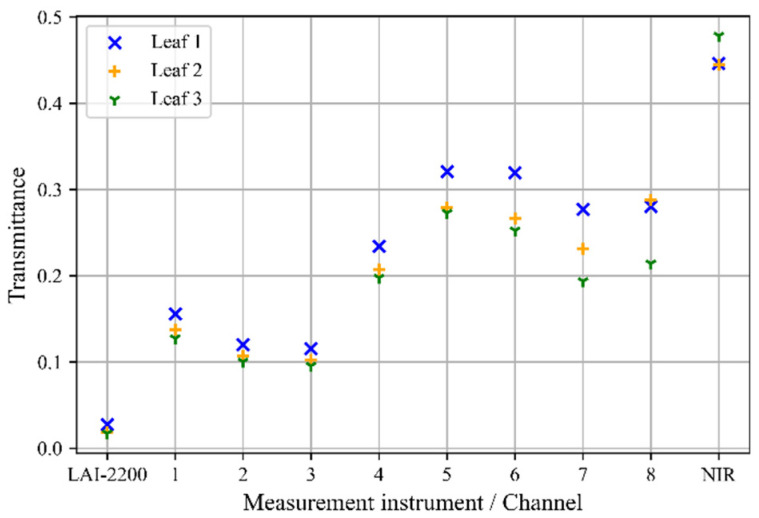
Comparison of the relative transmittance of multiple leaves of the Norway maple tree. The measurements were executed with the LAI-2200C and our sensor node channels 1 to 8 and NIR The plot shows the transmittance that was measured with the sensor node on the different channels in the visible range according to the datasheet (ams AG 2019) as well as in the NIR range.

**Figure 11 sensors-22-05048-f011:**
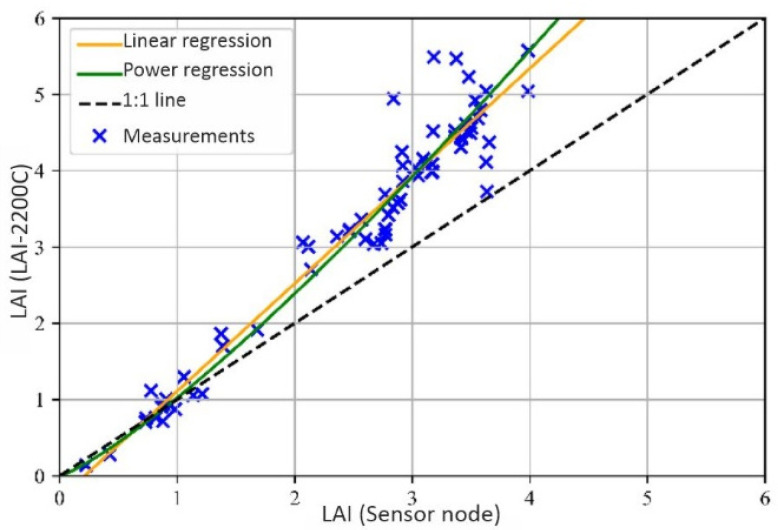
Relationship between the LAI that was measured with the LAI-2200C and with our sensor node in the field experiments. The plot also contains two optimized regression functions to fit the data: a linear and a power regression function.

**Figure 12 sensors-22-05048-f012:**
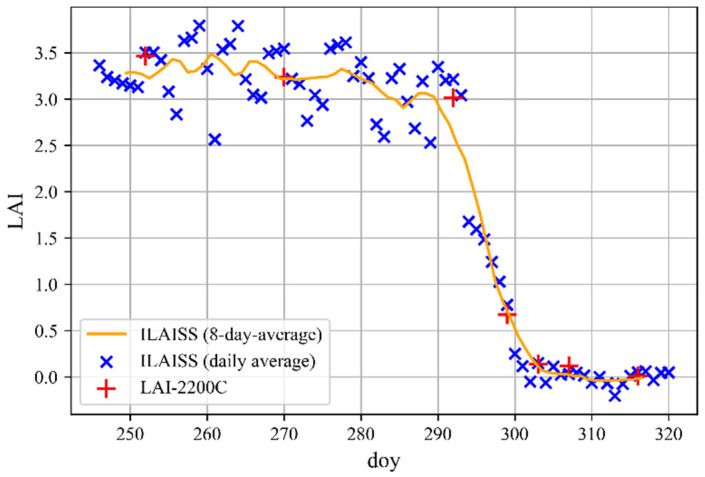
The LAI that was measured with the ILAISS and the LAI-2200C instrument during the 2021 campaign as a function of doy.

**Figure 13 sensors-22-05048-f013:**
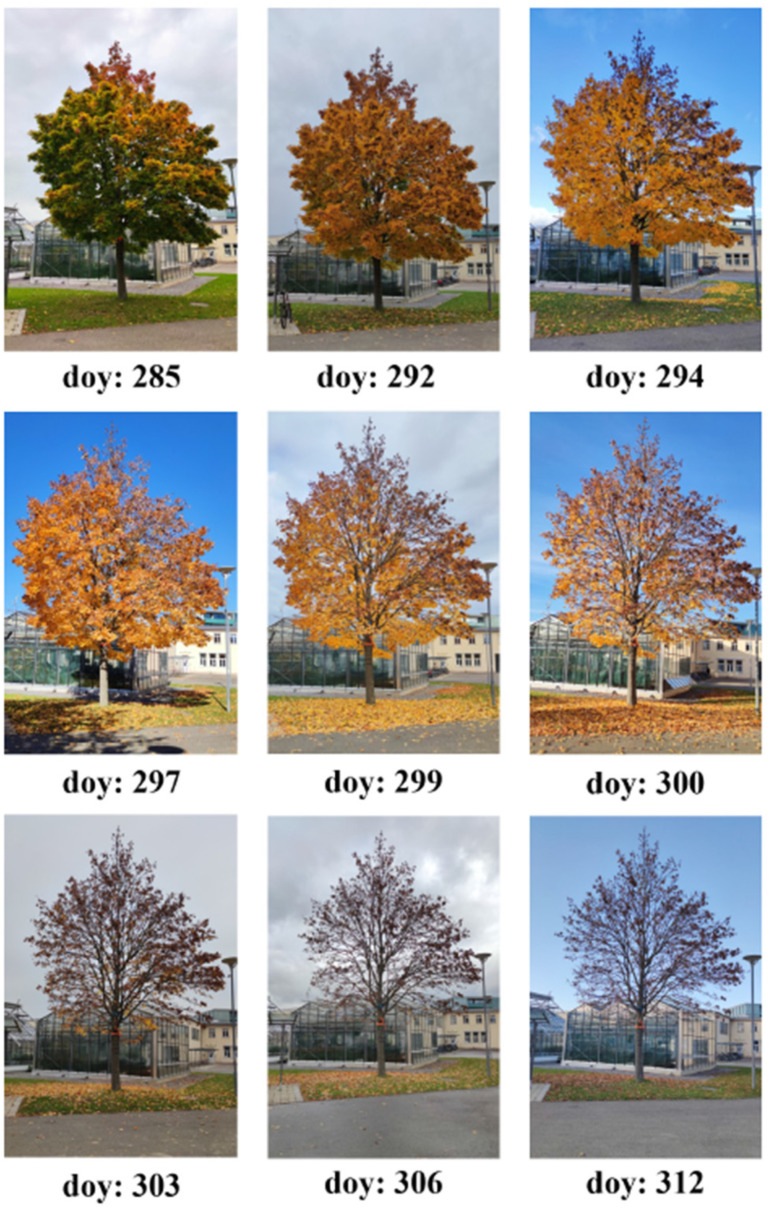
Photo series of the considered tree that were taken during the campaign.

**Table 1 sensors-22-05048-t001:** Positions and types of trees that were used in the field experiment.

Number	Tree Type	Position
1	Norway maple (Acer platanoides)	48.01357° N 8.83364° E
2	Bird cherry (Prunus avium)	48.01334° N 7.83457° E
3	Bird cherry(Prunus avium)	48.01368° N 7.83434° E

**Table 2 sensors-22-05048-t002:** Overview about the properties of the automatic LAI measurement systems and validation results that were achieved in indoor and outdoor deployments.

Publication	Measurement System Properties			Deployment	
	Name	Energy Supply	Spectral Response Range	Environment	Validation Results
[12]	Bauer	Battery	Blue spectrum	Crop land	$R^{2}=0.91;\;\mathrm{nRMSE}=0.07$
[14]	Shimojo	Battery	320–1100 nm	Greenhouse	$R^{2}=0.88;$ $R^{2}=0.97$
[16]	LAINet	Battery	Red spectrum	Coniferous forest	$R^{2}=0.81;\;\mathrm{RMSE}=0.34$
[15]	LAINet	Battery	Red spectrum	Crop land	$R^{2}=0.92;\;\mathrm{RMSE}=0.17$
This work	ILAISS	Energy Harvesting	Multiple (see Section 3.2)	Urban isolated trees	$R^{2}=0.997;\;\mathrm{RMSE}=0.149;\;\mathrm{nRMSE}=0.067$

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
