# Peer review of "Estimation of Leaf Area Index with a Multi-Channel Spectral Micro-Sensor for Wireless Sensing Networks"

_sensors, 2022, doi:10.3390/s22135048_

Round 1
Reviewer 1 Report
In this paper, the authors presented an approach to obtain the LAI with our developed distributed wireless sensor network based on the cheap and easy to integrate multichannel spectral sensor AS7341. With the described data process and filtering, the system achieves a very high accuracy with a nRMSE of only 0.098 when compared to the manually operated state-of-the-art instrument LAI-2200C. Long- term experiments reveal that the system is not only capable of continuous automated measurements with a high temporal resolution, but also enables correcting the LAI for the wooden components of trees. The novel points can be seen, and I think this paper can be published after some revision.
1)English in this paper needs improvement, which can make this paper more like a journal paper.
2)Figs.1 and 2 is not clear, and the authors can give a more details of sensor, especially, the 3D plot of structure.
3)The authors can do a short introduction about the future challenges in the devices in wireless sensor network. For instance, sensor, and algorithm.
4)Please polish the abstract. Please check the logic of abstract. Please add sentences to explain the meaning, the main points, the improvement and the promising application of the study. Plenty of detail data have given, however, in abstract, important procedures and results should be mentioned in simple manner. Please focus on the main points and the improvement of the study..
5)The authors can do a short introduction about the future development in the sensor technology. For instance,
Bao-Fei Wan, et al. IEEE Sensors Journal, vol. 21, no. 1, pp. 331-338, Jan.1, 2021, doi: 10.1109/JSEN.2020.3013289.
Author Response
Reviewer 1
In this paper, the authors presented an approach to obtain the LAI with our developed distributed wireless sensor network based on the cheap and easy to integrate multichannel spectral sensor AS7341. With the described data process and filtering, the system achieves very high accuracy with a nRMSE of only 0.098 when compared to the manually operated state-of-the-art instrument LAI-2200C. Long-term experiments reveal that the system is not only capable of continuous automated measurements with a high temporal resolution but also enables correcting the LAI for the wooden components of trees. The novel points can be seen, and I think this paper can be published after some revision.
1)English in this paper needs improvement, which can make this paper more like a journal paper.
Thank you for the suggestion. A native speaker checked the document to improve the language.
2)Figs.1 and 2 is not clear, and the authors can give a more details of sensor, especially, the 3D plot of structure.
Thank you for the comment. The following more detailed description of figures 1 and 2 has been added to paragraph 3.3:
„ The sensor chip AS7341 is mounted on a PCB as shown in Figure 2. A hole in the housing lid permits light to reach the sensor. This hole is covered by a diffuser disk of polymethyl methacrylate (PMMA), which ensures a homogenous illumination on the sensor. The housing of the node acts as a shadowing ring, blocking incoming radiation from an angle bigger than 180 degrees to prevent the sensing of radiation reflected by the ground. The glass dome protects the sensor and the circuitry from dirt or water. Figure 1 [17] shows a schematic of the cross-section of the sensing part of the sensor node. “
The added part is in yellow in the document and a comment identify the corresponding review and comment number. Additionally, we updated the labels in both figures and enlarged the font.
3)The authors can do a short introduction about the future challenges in the devices in wireless sensor network. For instance, sensor, and algorithm.
The following short introduction about the future challenges of the device, with focus on the wireless sensor network has been added in chapter 5:
“Challenging aspects, which will be addressed in the future work with focus on the wireless sensor network, will be prolongation of the sensor node’s operational lifetime through energy optimization and energy awareness. Algorithms will be developed to make the sensor node aware of its energy status and able to forecast its energy income in the near future. Based on its energy awareness, the sensor node will be able to actively tailor its functionality accordingly. To achieve this, the optical sensor will play an important role. It will simultaneously monitor the LAI and the energy income in terms of the input light. The advantage of this solution is that in comparison to algorithms using weather station data, we will have a very specific and local information to run optimized algorithms. Another relevant aspect, which will be further investigated, is the design of a special wireless mesh network for a forest environment.
Moreover, the integration of additional sensors on the node is planned. By embedding additional sensors, it will be possible to gather several types of data continuously and automatically in different levels of the crown, opening new possibilities in term of modeling of ecophysiological processes and stress tracing.”
The added part is marked in yellow in the document (line 550 - 570).
4)Please polish the abstract. Please check the logic of abstract. Please add sentences to explain the meaning, the main points, the improvement and the promising application of the study. Plenty of detail data have given, however, in abstract, important procedures and results should be mentioned in simple manner. Please focus on the main points and the improvement of the study.
Thank you for the suggestion. The abstract has been polished and the main points and the improvements obtained in this study pointed out with more emphasis.
The abstract now read:
“The Leaf Area Index (LAI) is a key parameter in the context of monitoring the development of tree crowns and plants in general. As parameters like carbon assimilation, environmental stress on carbon and the water fluxes within tree canopies are correlated to the leaves surface, this parameter is essential for understanding and modeling ecological processes. However, its continuous monitoring using manual state-of-the-art measurement instruments is still challenging. To address this challenge, we present an innovative sensor concept to obtain LAI based on the cheap and easy to integrate multi-channel spectral sensor AS7341. Additionally, we present a method for processing and filtering the gathered data, which enables very high accuracy measurements with a nRMSE of only 0.098, compared to the manually operated state-of-the-art instrument LAI-2200C (LiCor). The sensor embedded on a sensor node has been tested in long-term experiments, proving its suitability for continuous deployment over an entire season. It permits the estimation of both Plant Area Index (PAI) and Leaf Area Index (LAI) and provides the first wireless system that obtains the LAI solely powered by solar cells. Its energy autonomy and wireless connectivity make it suitable for a massive deployment over large areas and at different levels of the tree crown. It may be upgraded to allow the parallel measurement of Photo-synthetic Active Radiation (PAR) and light quality, relevant parameters for monitoring process-es within tree canopies.”
5)The authors can do a short introduction about the future development in the sensor technology.
The following short introduction about the future development in the sensor technology has been added:
“The developed sensor works ideally in the case of a homogenous tree crown, so a further challenge to be addressed in the sensor’s technical development is its optimization for non-homogenous crowns. This could be achieved by the use of sensors able to measure the incoming light at different angles, by using additional optics or by installing multiple regularly spaced sensors to measure LAI in correspondence with different distances from the trunk. Moreover, algorithms will be developed to enable the sensor to track the change of color of the leaves, for example, to identify a correlation with diseases or pests spread.”
The added part is in yellow in the document (line 572 - 578)

Reviewer 2 Report
In this paper, the authors proposed an innovative sensing concept to measure LAI by using the spectral sensor AS7341. Some suggestions are presented as follows.
1. In section 3, the authors present some methods of PAI estimation. The authors are suggested to identify any contribution in this section. If not, the existing work can be moved to the literature review.
2. In the system design, the authors are suggested to present the main idea of the design before presenting detail.
3. All equations should be reformatted.
4. Figure 11 shows that relation between the LAI measured with the LAI-2200C and with our sensor node in the 398 field experiments. The authors should explain more about this figure in detail.
Author Response
Reviewer 2
In this paper, the authors proposed an innovative sensing concept to measure LAI by using the spectral sensor AS7341. Some suggestions are presented as follows.
1) In section 3, the authors present some methods of PAI estimation. The authors are suggested to identify any contribution in this section. If not, the existing work can be moved to the literature review.
As suggested, I have moved the PAI estimation methods in the chapter Background theory for LAI estimation. In material and methods, the measurement methods developed in this study are now available.
2) In the system design, the authors are suggested to present the main idea of the design before presenting detail.
Thank you for your comment. As required, the following small introduction has been added at the beginning of chapter 3.3 to present the design concept:
„To estimate LAI an optical sensor has been developed based on the AS7341 (ams-OSRAM AG, Austria). The sensor is embedded in the energy-autonomous sensor node shown in Figure 2, which contains the electronics for data acquisition, processing and transmission “.
The added part is in yellow in the document (lines 159 - 161)
3) All equations should be reformatted.
Thank you for pointing it out. The equations were reformatted.
4) Figure 11 shows that relation between the LAI measured with the LAI-2200C and with our sensor node in the 398 field experiments. The authors should explain more about this figure in detail.
Thank you for your comment. As required, Figure 11 is explained more in detail:
„Figure 11 shows how both the linear and the power curve fit the 69 LAI measurements. First, we investigated the use of the linear function f(x)=mx+b plotted in orange in Figure 11. For m=1.409 and b=-0.299, the linear function is such that the MSE is minimized. With this polynomial function, a fitting error =0.094 and nRMSE = 0.113 is achieved. However, this function delivers negative output values for input values < 0.3 as shown in Figure 11 and, thus, is not well defined over the whole range.
An alternative function type that fulfills this requirement is given in equation 15 with a,r ∈R, plotted in green in Figure 11. For a=1.023 and r=1.223 the smallest MSE is achieved. In comparison to the linear regression, a smaller relative error is obtained, nRMSE = 0.117 is comparable. It follows that by applying this function for calibration, the deviation between the two instruments measurements is reduced approximately to one third. Moreover, the function is well defined over the whole LAI range. Figure 11 shows additionally in black dashes as reference the curve which would provide the ideal correspondence between the measurements executed with the newly developed sensor and with the LAI-2200C„
The added part is in yellow in the document (lines 413 - 428).
